

# Sedimentary archives of climate and sea-level changes during the Holocene in the Rhone prodelta (NW Mediterranean Sea)

Anne-Sophie Fanget[1], Maria-Angela Bassetti[1], Christophe Fontanier[2], Alina Tudryn[3], Serge Berné[1,2]

[1]Université de Perpignan Via Domitia, Centre de Formation et de Recherche sur les Environnements Méditerranéens (CEFREM), UMR 5110-CNRS, F-66860, Perpignan, France
[2]IFREMER, Laboratoire Environnements Sédimentaires, BP70, 29280 Plouzané, France
[3] GEOPS UMR 8148, Univ. Paris-Sud, CNRS, Université Paris-Saclay, Rue du Belvédère, Bât. 504-509, 91405 Orsay, France

*Correspondence to*: Anne-Sophie Fanget (annesophie.fanget@univ-perp.fr)

**Abstract.** A 7.38 m-long sediment core was collected from the eastern part of the Rhone prodelta (NW Mediterranean) at 67 m water depth. A multi-proxy study (sedimentary facies, benthic foraminifera and ostracods, clay mineralogy, and major elements from XRF) provides a multi-decadal to century-scale record of climate and sea-level changes during the Holocene. The early Holocene is marked by alternative silt and clay layers interpreted as distal tempestites deposited in a context of rising sea level. This interval contains shallow infra-littoral benthic meiofauna (e.g. *Pontocythere elongata*, *Elphidium* spp., *Quinqueloculina lata*) and formed between ca. 20 and 50 m water depth. The middle Holocene (ca. 8.3 to 4.5 ka cal. BP), is characterized, at the core site, by a period of sediment starvation (accumulation rate of ca. 0.01 cm yr$^{-1}$) resulting from the maximum landward shift of the shoreline and the Rhone outlet(s). From a sequence stratigraphic point of view, this condensed interval, about 35 cm-thick, is a Maximum Flooding Surface that can be identified on seismic profiles as the transition between delta retrogradation and delta progradation. It is marked by very distinct changes in all proxy records. Following the stabilization of the global sea level, the late Holocene is marked by the establishment of prodeltaic conditions at the core site, as shown by the lithofacies and by the presence of benthic meiofauna typical of the modern Rhone prodelta (e.g. *Valvulineria bradyana*, *Cassidulina carinata*, *Bulimina marginata*). Several periods of increased fluvial discharge are also emphasized by the presence of species commonly found in brackish and shallow water environments (e.g. *Leptocythere*). Some of these periods correspond to the multi-decadal to centennial late Holocene humid periods recognized in Europe (i.e. the 2.8 ka event and the Little Ice Age). Two other periods of increased runoffs at ca. 1.3 and 1.1 ka cal. BP are recognized, and are likely to reflect periods of regional climate deterioration that are observed in the Rhone watershed.
Keywords: Holocene. Delta. Benthic meiofauna. Sea level. Rapid climate changes. Hydrology.

## 1. Introduction

Deltas comprise a subaerial delta plain, where river processes dominate, a coarser-grained delta front where river and basinal processes interact, and a muddy submarine prodelta dominated by oceanic processes (Bhattacharya and Giosan, 2003;



Galloway, 1975; Postma, 1995). Most of the world's deltas were initiated during the early Holocene, between ca. 9.5 and 6 ka cal. BP, owing to a deceleration of sea-level rise (Stanley and Warne, 1994). They constitute key element of the continental margin system as they represent the first sink of sediments delivered by rivers (Trincardi et al., 2004).

Over the last decades, numerous studies have documented the land-sea evolution of these systems including the Amazon delta (Nittrouer et al., 1986), the Mekong delta (Ta et al., 2002; Xue et al., 2010), the Yellow delta (Liu et al., 2004a; Liu et al., 2007), and the Po delta (Amorosi et al., 2008; Cattaneo et al., 2003). The Rhone delta, one of the most important of the Mediterranean Sea, has also been widely investigated combining seismic, sedimentological and micropaleontological approaches (Boyer et al., 2005; Fanget et al., 2014; Fanget et al., 2013a; Fanget et al., 2013b; Gensous et al., 1993; Labaune et al., 2005).

In this paper, we study the sedimentary evolution of the Rhone prodelta during the last ca. 10.5 ka cal. BP as marked by changes in lithofacies and benthic meiofaunal assemblages (i.e. foraminifera and ostracods), in relation to the Holocene sea-level rise and climate changes. This study shows that (1) major phases of sea-level rise and delta evolution can be clearly identified based on several independent proxy records, and that (2) changes in fluvial discharge inferred, particularly, from ostracod assemblages in the upper part of the core are linked to the last major periods of rapid climate change of the Holocene (Mayewski et al., 2004; Wanner et al., 2014).

## 2. Regional geological and climatic setting

### 2.1. Geological history of the Rhone subaqueous delta

In the Gulf of Lions (NW Mediterranean), the Rhone delta (Fig. 1) occupies a deeply incised Messinian valley infilled with thick (~2 km) Plio-Quaternary sediments (Lofi et al., 2003), mainly delivered by the Rhone River (Aloïsi et al., 1977). For the last ca. 500 ka, borehole data demonstrated that shelf deposits are primarily made-up of forced-regressed sequences formed in response to 100-kyr glacio-eustatic cycles (Bassetti et al., 2008; Frigola et al., 2012; Sierro et al., 2009). These authors also demonstrated that higher frequency cycles, as well as sub-orbital climate changes, were nicely recorded within paleo-prodeltaic sedimentary archives. Following the Last Glacial Maximum (LGM, ca. 21 ka cal. BP; Mix et al., 2001), rapid sea-level rise led to the retrogradation of Rhone delta, and formation of a wedge of transgressive (backstepping) deposits thickening landward. The most prominent feature is an elongated paleo-deltaic complex, named the Early Rhone Deltaic Complex (Berné et al., 2007), and formed during the Younger Dryas and the Preboreal (Fig. 1). After ca. 7 ka, stabilization of sea level allowed the progradation of a series of regressive deltaic lobes (Fanget et al., 2014), corresponding to the overall eastward migration of the Rhone distributaries. In total, these transgressive and regressive deposits form the Rhone subaqueous delta that reaches, along the modern delta front, up to 50 m in thickness, and pinches out at a present water depth of ca. 90 m (Gensous and Tesson, 1997).

The early Holocene deposits (called seismic unit U500), which rest on a wave ravinement surface (called D500), formed a transgressive parasequence (Labaune et al., 2005) made of tempestite deposits (Fanget et al., 2014). They are separated from



middle and late Holocene deposits by a Maximum Flooding Surface (MFS, called D600), which corresponds to a condensed interval. The age of this surface varies along-strike between ca. 8 and 3 ka cal. BP, and reflects, at a given site, the duration

of condensation and/or erosion (Fanget et al., 2014). After the stabilization of global sea level (ca.7 ka cal. BP), the middle and late Holocene Rhone outlets shifted progressively eastward, under natural and/or anthropic influence. As a result, several deltaic lobes are formed (Fig. 1) (Arnaud-Fassetta, 1998; L'Homer et al., 1981; Provansal et al., 2003; Rey et al., 2005; Vella et al., 2008; Vella et al., 2005). Saint Ferréol, which is related to the "Rhône de Saint Ferréol" Channel, is the first and largest paleo-deltaic lobe. It started to prograde around 7 ka cal. BP (L'Homer et al., 1981). The Ulmet lobe, located eastward

and linked to the "Rhône d'Ulmet" Channel formed simultaneously to the Saint Ferréol lobe. Westward of the Saint Ferréol lobe, the Peccais lobe, related to the "Rhône de Peccais" Channel, appeared to be posterior to the erosion of the St Ferréol (Rey et al., 2005; Vella et al., 2005). During the Little Ice Age, the Bras de Fer lobe, linked to the "Rhône de Bras de Fer" Channel, formed between 1587 and 1711 AD (Arnaud-Fassetta, 1998). Until 1650 AD, the "Rhône de Bras de Fer" Channel is considered as synchronous to the "Rhône du Grand Passon" Channel (Arnaud-Fassetta, 1998). The "Rhône de Bras de

Fer" Channel shifted to the east up to the present-day position of the Grand Rhone River after several severe floods that occurred in 1709-1711 AD. The progradation of these lobes is primarily influenced by changes of sediment fluxes (Arnaud-Fassetta, 2002; Bruneton et al., 2001; Provansal et al., 2003), and thus by climate.

## 2.2. Holocene climate and its regional characteristics

High fluctuations in rainfall and low-amplitude temperature variations are observed during the Holocene (Davis et al., 2003;

Mayewski et al., 2004; Seppä et al., 2009; Wanner et al., 2008; Wanner et al., 2011). Examination of globally distributed paleoclimate records led to identify 8 to 10 multi-decadal to century-scale cooling events interrupting periods of relatively stable and warmer climate (Mayewski et al., 2004; Wanner et al., 2008; Wanner et al., 2014; Wanner et al., 2011). These periods are known as Rapid Climate Changes (RCC; Mayewski et al., 2004) or Cold Relapses (CR; Bassetti et al., 2016; Jalali et al., 2016; Wanner et al., 2014). The 8.2 ka cal. BP cold event (CR0, Table 1) occurred during the early Holocene

(Barber et al., 1999), a period of progressive warming that induced ice cap melting and freshwater outbursts to the oceans from North American glacial lakes.

During the warm middle Holocene, the most significant events in terms of temperature occurred at 6.4, 5.3 and 4.2 ka cal. BP (CR1, CR2 and CR3, respectively; Table 1) (Walker et al., 2012; Wanner et al., 2014). The 4.2 ka event, that may have played a role in the collapse of various civilizations (Magny et al., 2013), is characterized by increased drought in

North America, Asia and South Mediterranean region.

During the cooler late Holocene, the 2.8 ka cal. BP cold relapse (CR4, Table 1) might be responsible for the decline of the Late Bronze Age civilization (Do Carmo and Sanguinetti, 1999; Weiss, 1982), and CR5 (Table 1) matches the Migration Period that occurred around 1.4 ka cal. BP (Wanner et al., 2014). The late Holocene cooling trend culminated during the Little Ice Age (LIA, CR6 in Table 1) between the 13[th] and 19[th] centuries (Frezza and Carboni, 2009).





In the Rhone catchment area, these CR (or at least CR0 and CR6) are marked by increased river runoff. CR0, or the so-called 8.2 ka event (Alley et al., 1997), is indeed marked by high lake levels due to more intense rainfall (Magny and Begeot, 2004; Magny et al., 2003). The LIA (CR6), which is well-documented in the Rhone watershed, is characterized by a period of Alpine glacier advance (Goehring et al., 2011; Ivy-Ochs et al., 2009), high lake levels (Magny et al., 2010), and marked increase of Rhone River floods (Pichard, 1995; Pichard and Roucaute, 2014).

## 3. Material and Methods

The present paper is based on a multi-proxy analysis of a 7.38 m-long piston core (RHS-KS55) collected in front of the paleo-"Rhône de Bras de Fer" and "Grand Passon" Channels at 67 m water depth (latitude = N 43°14.35'; longitude = E 4°40.96') during the RHOSOS cruise (*R/V "Le Suroît"*, September 2008).

The detailed architecture of the Rhone subaqueous delta was determined from high-resolution seismic data (2000-5200 Hz
Chirp system), together with core data constrained by $^{14}$C dates (Fanget et al., 2014). Radiometric dates were measured with accelerator mass spectrometer (AMS) $^{14}$C on well-preserved benthic foraminifera or shells at Poznan Radiocarbon Laboratory (PRL, Poland), and at *Laboratoire de Mesure 14C* (LMC14) at *Commissariat à l'Energie Atomique* (CEA, France). Nevertheless, due to the low quantity of biogenic material in the proximal part of the Rhone prodelta, we experienced difficulties in dating core RHS-KS55 and observed some age inversions. Based on seismic and lithofacies
correlations at the regional scale (Fanget et al., 2014), we excluded some $^{14}$C dates for core RHS-KS55. As a result, eight robust $^{14}$C dates were used to create the age model for the last ca. 9 kyr cal. BP. Ages are $\delta^{13}$C-normalised conventional $^{14}$C years, and are corrected for an assumed air-sea reservoir effect of 400 years. Calendar ages were calculated using the program CLAM (version 2.2, Blaauw, 2010) and the Marine 13 calibration curve (Reimer et al., 2013). The age model was based on the linear interpolation between the dated levels using basic (non-Bayesian) age-depth modeling software (Blaauw,
115      2010).

Core RHS-KS55 was split lengthwise, photographed, and visually described in order to identify sedimentological facies. Core RHS-KS55 was then analyzed using an Avaatech XRF Core Scanner (Richter et al., 2006) at IFREMER (Brest, France). Semi-quantitative analyses of major and minor elements were performed by scanning the surface of split sediment cores with a sampling step of 1 cm and a counting time of 20 s. Two runs with X-ray source voltages and intensities of 10
kV-200 µA and 30 kV-1000 µA were carried out. Only data for Titanium (Ti) element that is commonly related to terrigenous-siliciclastic components in the sediment (Richter et al., 2006) are reported in the present study.

Core RHS-KS55 was then subsampled for benthic microfaunal analyses (i.e. ostracods and foraminifera) and clay mineralogy. Three-cm thick slides were collected using a sampling interval of 10 cm through the core, except from the base of the core to ca. 500 cm, where thick slides and sampling step were slightly modified. A total of 79 samples were washed
over a 63 µm sieve and the residues were dried and dry-sieved using a 125 and 150 µm mesh screens. Ostracods and benthic foraminifera were hand-sorted from the >125 and >150 µm fractions, respectively, and stored in Plummer slides. To illustrate



the diversity of benthic meiofauna, total abundance (values normalized for a 100 cm$^3$ sample volume), species richness ($S$), Shannon index ($H$), and Evenness index ($E$) (Hayek and Buzas, 1997; Shannon, 1948) were calculated, as described in Murray (2006), for each level. To highlight vertical patterns in benthic meiofaunal communities, hierarchical clustering was

also performed on the 79 samples and the 16 major species (i.e. occurring with more than 5% in at least one sample) by mean of the PAST$^©$ software (version 2.09, 2011; Hammer et al., 2001). Cluster analyses were based on the arcsinus values of the square root "$pi$", where "$pi$" is the relative abundance (%) of the species $i$ divided by 100. A tree diagram was constructed according to the Ward's method based on the squared Euclidean distances.

Fraction inferior to 63 μm was used to perform clay minerals analyses. X-ray diffraction (XRD) on oriented mounts of non-

calcareous clay-sized (< 2 μm) particles was conducted to identify clay minerals with the PANalytical diffractometer, following the routine of the GEOPS Laboratory (Paris Sud University, France) (Liu et al., 2004b; Liu et al., 2008). Three XRD runs were carried out, following air-drying, ethylene–glycol solvation during 24 hours, and heating at 490 °C during 2 hours. Position of the (001) series of basal reflections on the three XRD diagrams was used to identify clay minerals. Semi-quantitative estimates of peak areas of the basal reflections for the main clay mineral assemblages of illite (10 Å), smectite

(including mixed-layers) (15–17 Å), and kaolinite/chlorite (7 Å) were performed on the glycolated curve using the MacDiff software (Petschick, 2000). Relative proportions of kaolinite and chlorite were determined using the ratio 3.57/3.54 Å of the peak areas.

## 4. Results

### 4.1. Seismic stratigraphic framework, age model and sedimentation rates

Seismic discontinuities and seismic units described in the following section are based on Fanget et al. (2014). Several Deglacial and Holocene seismic units bounded by well-marked discontinuities are identified at the core site (Fig. 2).

Seismic data highlights that core RHS-KS55 goes through surface D500, and represents an expanded record of seismic units U500, U600, and U610. Seismic unit U620a is missing in the studied core (Fig. 2).

The age of seismic unit U500, that corresponds to the early Holocene transgressive parasequence (Fanget et al., 2014), is

poorly constrained in core RHS-KS55. Considering the age of the underlying deposits of seismic unit U400 in this area (i.e. paleo-deltaic complex of the Rhone (ERDC), ca. 10.5 ka cal. BP, Berné et al., 2007), we assume that $^{14}$C dates obtained in this unit are generally biased, because of reworking occurring in shallow water environment. Only the uppermost part of this unit is confidently dated between ca. 9.2 and 8.3 ka cal. BP (Table 2); the older ages obtained within seismic unit U500 are possibly the result of reworking during the transgression of an underlying Younger Dryas/Preboreal delta front. The upper

boundary of seismic unit U500, called D600 and interpreted as a Maximum Flooding Surface (MFS, Fanget et al., 2014), corresponds to a condensed interval formed between ca. 8.3 and 4.5 ka cal. BP in the studied core (Fig. 3). It is characterized by very low sedimentation rate of ca. 0.01 cm yr$^{-1}$ (Fig. 4). Seismic unit U600 progrades on D600, and is related to the marine component of the St Ferréol lobe and Ulmet lobe (Fanget et al., 2014). $^{14}$C dates indicate that seismic unit U600 was





deposited between ca. 4.5 and 0.9 ka cal. BP (Fig. 3). Sedimentation rates through this interval oscillated between 0.03 and 0.4 cm yr$^{-1}$ (Fig. 4). Highest sedimentation rates are recorded along the uppermost part of this unit (i.e. between 300 and 110 cm, Fig. 4). Finally, seismic unit U610, which corresponds to the activity of the Grand Passon and Bras de Fer channels, was formed between ca. 900 and 280 a cal. BP (Fig. 3) (Fanget et al., 2014). As seismic unit U620a is missing within core RHS-KS55, we estimate that the top of the core has an age of ca. 280 a cal. BP. Sedimentation rate through seismic unit U610 is estimated at ca. 0.11 cm yr$^{-1}$ (Fig. 4).

## 4.2. Sedimentary features

Based on lithological description (including lithofacies, sedimentary structures, bioturbation and color) of core RHS-KS55 (Fanget et al., 2014), three main sedimentary facies (i.e. Facies 1, 2 and 3 (including 3a and 3b) are identified and summarized as follows:

Facies 1: From 738 (core bottom) to 460 cm, core RHS-KS55 consists of numerous silt or very fine sand laminae (in the sense of Campbell, 1967) interlaminated with grayish and beige silty clay with millimeters to several centimeters spacing (Fig.4). Within these very thin laminae (mm to few cm-thick), which are characterized by erosional basal contacts, no sedimentary structures can be identified.

Facies 2: From 460 to 430 cm, a peculiar interval consisting of heterolithic content in a grayish silty clay matrix is observed (Fig. 4). *Turritella* sp., as well as several bivalves (e.g. *Acanthocardia echinata*, *Arca tetragona*, *Nucula* sp.) and bryozoans are identified.

Facies 3a: From 430 to 320 cm, sediment consists of beige silty clay without visible sedimentary structures. Diffuse veneers of yellowish lighter levels and spot of darker sediments (richer in hydrotroilite), clearly obliterated by intense bioturbation, are observed (Fig. 4). Scattered bryozoans debris, *Turritella* sp., and bivalves are encountered in this interval.

Facies 3b: From 320 to 0 cm, sediment consists of grayish and beige silty clay and contains abundant hydrotroilites and bioturbation.

## 4.3. Clay mineralogy and XRF data

Clay mineral assemblages are dominantly composed of illite, with values ranging from ca. 55 to 85% (Fig. 5). Illite content exhibits high and quite constant values (ca. 70%) from the bottom of the core to 465 cm. From 455 to 360 cm, illite content decreases to ca. 60%. A strong increase is observed at 340 cm with values reaching ca. 80%. Illite content remains high between 340 and 110 cm. At 110 cm, illite content drops to ca. 55% and is generally lower from 100 to 30 cm. Along the uppermost 30 cm of the core, a progressive increase in illite content is identified. Smectite content is inversely correlated to illite content throughout core RHS-KS55 (Fig. 5). It has very low values (ca. 2.5%) from 738 to 465 cm, and exhibits higher (from ca. 5 to 25%) and erratic values along the uppermost 460 cm of the core. Chlorite content reaches ca. 20% in the lower part of the core (from the bottom to 350 cm), and drops near 0% between 350 and 120 cm (Fig. 5). Chlorite content increases to ca. 20% along the uppermost 120 cm of the core. Kaolinite content is very low throughout core RHS-KS55 with values



close to 0% from the bottom of the core to 360 cm, and ranging from ca. 7 to 10% between 360 cm and the top of the core (Fig. 5).

Major changes in clay mineral assemblages occur simultaneously with changes in sedimentary facies (Fig. 5).

Measurements of the bulk intensity of Ti (Fig. 5) show erratic values from the bottom of the core to ca. 460 cm. Ti content increases progressively from ca. 460 to 400 cm and is characterized by high values between 400 and 20 cm. Within this interval, some little oscillations in Ti content are observed. From 20 cm to the top of the core, Ti content decreases strongly.

## 4.4. Ostracod fauna

### 4.4.1. Ostracod density and diversity indices

The number of ostracods per sample varies greatly from 68 to 12 821 ind./100 cm$^3$ in core RHS-KS55 (Fig. 6). From the base of the core to 482 cm, density ranges from 68 to 1 266 ind./100 cm$^3$. The lowest values are observed from 642 to 632 cm, whereas four peaks of 1 141, 1 243, 1 006, and 1 266 ind./100 cm$^3$ are identified at 702, 622, 571, and 541 cm, respectively. From 471 to 352 cm, the number of counted ostracods increases strongly with values reaching up to 12 821 ind./100 cm$^3$ and 10 539 ind./100 cm$^3$ at 432 and 412 cm, respectively. The number of ostracods per sample drops to 222 ind./100 cm$^3$ at 332 cm, and ranges from 305 to 1182 ind./100 cm$^3$ between 322 and 162 cm. Density decreases between 152 and 82 cm, and then increases progressively along the uppermost 82 cm of the core.

Species richness ($S$) oscillates between 7 and 31 species per sample through the core, and follows approximately the same trend that density (Fig. 6). Lowest values of $S$ are recorded when ostracods abundances are minimal, i.e. at 642 and 632 cm, and between 152 and 82 cm. $S$ is maximal within the interval comprising between 482 and 442 cm. The Shannon index ($H$) varies between 1.2 and 3.0 through the core (Fig. 6). $H$ increases progressively from the base of the core to 492 cm. $H$ reaches maximal values (ca. 3.0) between 482 and 442 cm, and decreases progressively from 433 to 82 cm. Along the uppermost 82 cm of the core, $H$ gradually increases (from ca. 1.4 to 2.2). The Evenness index ($E$), which is comprised between 0.2 and 0.7, follows approximately the same trend that $H$ (Fig. 4). The lowest values of $E$ are observed from 738 to 492 cm, whereas the maximal values are observed from 482 to 422 cm. $E$ is relatively constant along the uppermost 412 cm of the core, with values oscillating around 0.5.

### 4.4.2. Cluster analysis

R-mode cluster analysis allows us to identify six ostracod clusters plus one single species when a cut-off level of 1.4 is applied (Figs. 7 and 8).

Cluster A is made of *Semicytherura incongruens*, *Pontocythere elongata*, and *Semicytherura* sp. It has a maximal contribution from 738 to 512 cm, with erratic values ranging from ca. 5 to 25%. It decreases strongly from 512 to 442 cm, and disappears completely along the uppermost 432 cm of the core.



Cluster B is composed of *Propontocypris pirifera*, *Cytherissa* sp., *Eucythere* sp., *Aurila* sp., and *Cytheridea neapolitana*. It shows a low contribution through the core, with values generally <10%. It increases only between 482 and 442 cm, where a maximal value of 25% is reached.

Cluster C is constituted by *Cytherella* sp., *Cytheropteron alatum*, *Cytheropteron monoceros*, and *Carinocythereis carinata*. It exhibits a low contribution (<10%) from 738 to 492 cm. It increases strongly between 492 and 372 cm, to reach up to ca. 40% of the ostracod fauna at 432 cm. Cluster C decreases progressively, and has a minimal contribution along the uppermost 362 cm of the core.

The single species corresponds to *Leptocythere* spp. This species dominates ostracod fauna from 738 to 492 cm, with values ranging from ca. 41 to 73%. Along the uppermost 492 cm of the core, *Leptocythere* spp. oscillates between low (<5%) and high (ca. 30-40%) values. The lowest contributions of this species are recorded from 472 to 432 cm, 332 to 302 cm, 232 to 182 cm, and 122 to 82 cm.

Cluster D is made of *Cytheropteron rotundatum* and *Krithe* spp. (juvenile *Krithe* and *K. pernoides*). From 738 to 362 cm, it has a minimal contribution (<5%). It increases strongly along the uppermost 352 cm of the core, with values ranging from ca. 30 to 60%, and reaches a peak of ca. 77% at 92 and 82 cm.

Cluster E is composed of *Argilloecia* spp., *Loxoconcha laevis*, and *Paradoxostoma* sp. It exhibits a moderate contribution through the core, with values oscillating generally between ca. 5 and 15%. However, four peaks of ca. 37, 45, 35, and 27% are recorded at 492, 342-332, 112, and 72 cm, respectively.

Cluster F is constituted by *Sagmatocythere* sp., *Bosquetina dentata*, and *Pterigocythereis jonesii*. It shows a minimal contribution (<5%) from 738 to 492 cm. It increases between 482 and 162 cm, with values ranging from ca. 20 to 35%, except between 352 and 332 cm, where a decrease (<15%) is observed. From 162 to 72 cm, Cluster F exhibits erratic values oscillating between ca. 2 and 20%. Along the uppermost 72 cm of the core, it increases slightly.

### 4.5. Benthic foraminiferal fauna

### 4.5.1. Benthic foraminiferal density and diversity indices

The number of benthic foraminifera per sample varies greatly from 404 to 74 642 ind./100 cm$^3$ through the core (Fig. 6). From 738 to 482 cm, density shows erratic values oscillating between 404 and 7 130 ind./100 cm$^3$, and increases progressively. The number of counted specimens increases strongly between 472 and 382 cm, with values ranging from 12 386 to 74 642 ind./100 cm$^3$. From 382 to 352 cm, density decreases rapidly, and drops to 2 995 ind./100 cm$^3$. Along the uppermost 342 cm of the core, benthic foraminiferal densities remain quite constant and below 3 000 ind./100 cm$^3$.

Species richness ($S$) oscillates between 13 and 49 species per sample through the core (Fig. 6). The lowest values of $S$ (from 13 to 26) are recorded from 738 to 632 cm. $S$ increases from 622 to 492 cm, except between 542 and 532 cm where a decreased is observed. The highest values of $S$ are encountered from 482 to 212 cm, with values oscillating between 34 and 49 species per sample. $S$ slightly decreases along the uppermost 202 cm of the core and remains quite constant. The Shannon





index (*H*) varies between 1.2 and 3.2 through the core RHS-KS55 (Fig. 6). From 738 to 642 cm, *H* decreases progressively from 2.3 to 1.2. Erratic values, oscillating between 1.6 and 2.7, are observed from 632 to 492 cm. Even if *H* slightly

decreased between 452 and 402cm, the highest values are recorded between 482 and 352 cm. From 342 to 132 cm, *H* decreases progressively, whereas it increases slightly along the uppermost 122 cm of the core. The Evenness index (*E*) exhibits relatively low values through the core, ranging from 0.2 to 0.5, and follows exactly the same trend that *H* (Fig. 6).

### 4.5.2. Cluster analysis

R-mode cluster analysis allows us to distinguish four benthic foraminiferal clusters when a cut-off level of 2.4 is applied

(Figs. 7 and 9).

Cluster 1 is composed of *Elphidium* spp. (including *E. advenum*, *E. crispum*, *E. decipiens*, *E. granosum*, *E. incertum*, *E. macellum*, and *E. margaritaceum*), *Nonionella turgida*, and *Quinqueloculina lata*. From 738 to 492 cm, Cluster 1 dominates strongly with abundances oscillating between ca. 58 and 91%. The contribution of Cluster 1 drops to ca. 15% at 472 cm. It has a minimal contribution along the uppermost 472 cm of the core, and especially between 342 and 62 cm, where it exhibits

values <10%.

Cluster 2 is constituted by *Cassidulina carinata*, *Bulimina marginata*, and *Valvulineria bradyana*. It is characterized by a low contribution (less than 10%) from 738 to 492 cm. It increases progressively from ca. 15 to 72% between 482 and 202 cm, and remains quite constant along the uppermost 192 cm of the core, with values oscillating between ca. 50 and 72%.

Cluster 3 is made of *Haynesina depressula*, *Ammonia beccarii*, and *Eggerella scabra*.

It has a low contribution (<15%) through the core RHS-KS55. From 738 to 492 cm, it exhibits relatively erratic values, and three peaks of ca. 12, 5, and 6% are observed at 699, 632, and 532 cm, respectively. Cluster 3 has a minimal contribution (<1%) between 482 and 342 cm, and increases slightly from 332 to 162 cm. From 162 to 82 cm, it increases progressively to reach a value of ca. 9%. Cluster 3 decreases again between 82 and 32 cm, and increases slightly along the uppermost 32 cm of the core.

Cluster 4 is constituted by *Pseudoeponides falsobeccarii*, *Textularia agglutinans*, *Hyalinea balthica*, *Melonis barleeanus*, *Bulimina aculeata*, *Sigmoilopsis schlumbergeri*, and *Cibicides lobatulus*. It shows a low contribution from 738 to 492 cm, with erratic values ranging from ca. 5 to 22%. It increases strongly between 482 and 402 cm, where values of ca. 55% are reached. From 402 to 292 cm, Cluster 4 decreases progressively. It remains quite constant along the uppermost 292 cm of the core, with values oscillating between ca. 13 and 25%.

## 5. Discussion

### 5.1. Record of Holocene sea-level rise and Rhone delta evolution

Seismic stratigraphy, sedimentological (including clay minerals) and benthic meiofauna data described in the previous section allow the sub-division of the studied core into three main intervals. These intervals fairly match the tripartite division



of the Holocene (Walker et al., 2012; Wanner et al., 2014), and are closely linked to the Holocene sea-level history, and to
the Rhone deltaic system evolution.

### 5.1.1. Interval 1 (ca. 10.5-8.3 ka cal. BP)

This interval encompasses most of the early Holocene. The age of the bottom of the core up to ca. 460 cm cannot be dated
precisely because it corresponds to a transgressive parasequence (seismic unit U500, Fig. 2), that formed in a context of
shallow-marine environment. Based on the age of the underlying deposits (i.e. the ERDC, seismic unit U400) and on $^{14}$C
dates, this interval was deposited between ca. 10.5 and 8.3 ka cal. BP (i.e. the early Holocene; Walker et al., 2012).
Tempestite (storm-induced) deposits, which are commonly formed in lower to middle shoreface environments during
periods of storm decelerating flows (Myrow, 1992; Myrow and Southard, 1996; Pérez-López and Pérez-Valera, 2012),
characterize this interval (Fig. 4). The intercalation of fine clay and silt layers (corresponding to strong variations in Ti
content, Fig. 5) suggests that these deposits are distal tempestites (i.e. turbidite-like deposited below the storm wave base;
Myrow, 1992; Pérez-López and Pérez-Valera, 2012). This facies is interpreted to correspond to an hydrodynamic regime
resulting from the combination of E-SE storm waves and flood events (i.e. 'wet storms' of Guillén et al., 2006), which
regularly winnow the seafloor (Fanget et al., 2014). Tempestite deposits mainly contain foraminifera belonging to Cluster 1
(*Elphidium* spp., *N. turgida*, *Q. lata*), and ostracods belonging to Cluster A (*S. incongruens*, *P. elongata*, *Semicytherura* sp.),
and to the genus *Leptocythere* (Figs. 8 and 9). Benthic foraminiferal species, like *N. turgida*, are typical of shallow prodeltaic
environment enriched in organic matter of continental origin (e.g. Barmawidjaja et al., 1992; De Rijk et al., 2000; Diz and
Francés, 2008; Van der Zwaan and Jorissen, 1991). *Elphidium* spp. and *Q. lata* are commonly reported in sandy silty
substrates subject to strong hydrodynamic processes (e.g. Donnici and Serandrei Barbero, 2002; Jorissen, 1988; Rossi and
Vaiani, 2008; Sgarrella and Moncharmont Zei, 1993). Similar observations are described in the modern Rhone subaqueous
delta (Goineau et al., 2011; Goineau et al., 2015; Mojtahid et al., 2009). Similarly, ostracods content of Cluster A are
represented by littoral to sublittoral/phytal marine forms (e.g. Bonaduce et al., 1975; Cabral et al., 2006; Carbonel, 1980;
Peypouquet and Nachite, 1984; Zaïbi et al., 2012). The genus *Leptocythere* is commonly found in brackish and shallow
water environments, and many *Leptocythere* are known to be euryhaline species (e.g. Anadon et al., 2002; Boomer and
Eisenhauer, 2002; Carbonel, 1973, 1980; Frenzel and Boomer, 2005; Gliozzi et al., 2005; Van Morkhoven, 1963).

According to paleoenvironmental reconstruction based on benthic meiofauna from core RHS-KS55, the early Holocene is
characterized, in the Rhone subaqueous delta, by high energy hydrodynamic processes and significant organic matter input
of continental origin typical of shallow infra-littoral setting. This interpretation is in agreement with the occurrence of
tempestite deposits, and the global estimates of sea-level rise during the early Holocene (e.g. Bard et al., 1996; Fairbanks,
1989; Smith et al., 2011). Based on the sea-level curve of Stanford et al. (2011), the base of the core (estimated at. ca.
10.5 ka cal. BP) corresponds to a sea level of ca. 50 m below its present-day position. Due to the location of core RHS-KS55
at a water depth of 67 m and its length of 7.38 m, a paleo-water depth of ca. 24 m can be estimated at the base of the core
(subsidence and compaction being considered as negligible). At the top of the tempestite facies (i.e. at ca. 460 cm), which is



dated at ca. 9.2 ka cal. BP, a paleo-water depth of ca. 52 m is estimated. Thus, the resulting rate of sea-level rise within this interval (738-460 cm) is ca. 20 mm yr$^{-1}$. This value matches the one found by Stanford et al. (2011) for the early Holocene. Thus, we consider that tempestite deposits, preserved within this transgressive interval (seismic unit U500), are formed at

water depth ranging from ca. 20 to 50 m. In the Rhone subaqueous delta, we consider the tempestite facies as a relatively good paleo-bathymetric marker and we have been able to correlate it over a large prodelta area (see core RHS-KS40, RHS-KS22, and RHS-KS39 in Fanget et al. (2014)).

### 5.1.2. Interval 2 (ca. 8.3-4.5 ka cal. BP)

The interval comprised between 460 and 430 cm corresponds to a period ranging from ca. 8.3 to 4.5 ka cal. BP (i.e. the
middle Holocene, Fig. 3), when the Rhone outlet(s) was situated 10 to 30 km landward from the modern shoreline. Considering the resolution of our seismic data (in the order of ca. 0.5 m), it corresponds to the position of the MFS (surface D600, Fig. 2) that marks the transition between retrogradation and progradation. Very low sediment accumulation (ca. 0.01 cm yr$^{-1}$, Fig., 4), abundant shell concentration, and very rich microfossil content (up to ~13 000 ostracods/100 cm$^3$ and ~75 000 foraminifera/100 cm$^3$, Fig. 6) indicate sediment starvation and condensation within this interval which separates
transgressive (below) from regressive (above) deposits. It consists in a silty clay matrix incorporating coarse-grained sediments with reworked shoreface material and shell hash. Benthic foraminifera belonging to Cluster 4 (*P. falsobeccarii*, *T. agglutinans*, *H. balthica*, *M. barleeanus*, *B. aculeata*, *S. schlumbergeri*, *C. lobatulus*), and ostracods belonging to Cluster B (*P. pirifera*, *Cytherissa* sp., *Eucythere* sp., *Aurila* sp., and *C. neapolitana)*, Cluster C (*Cytherella* sp., *C. alatum*, *C. monoceros*, and *C. carinata*), and Cluster F (*Sagmatocythere* sp., *B. dentata*, and *P. jonesii*) are dominant within this interval
(Figs. 8 and 9). Except for *C. lobatulus* which is preferentially found in high energy shallow-water setting (e.g. Bartels-Jónsdóttir et al., 2006; Javaux and Scott, 2003; Milker et al., 2011; Murray, 2006), foraminifera assemblage is mainly composed of species thriving under stable environment characterized by marine-derived organic matter supplies and well-oxygenated sediments (e.g. De Rijk et al., 2000; Debenay and Redois, 1997; Fontanier et al., 2008; Goineau, 2011; Goineau et al., 2011; Goineau et al., 2015; Mendes et al., 2004; Mojtahid et al., 2009). Clusters B and F are mainly composed of
shallow infra-littoral ostracods (e.g. Bonaduce et al., 1975; Frenzel and Boomer, 2005; Guernet et al., 2003; Ruiz et al., 1997; Zaïbi et al., 2012), whereas Cluster C is primarily made of circa-littoral and epi-bathyal species (e.g. Bonaduce et al., 1975; El Hmaidi et al., 2010; Yamaguchi and Norris, 2012).

The middle Holocene condensed section is very well identified thanks to benthic microfossils indicating mixed assemblages belonging to diverse environments, from infra-littoral to epi-bathyal settings. Shallow-water species highlight incorporation
of the previous shoreface and delta mouth sediments that were left in situ during the transgressive submersion. Circa-littoral and epi-bathyal species indicate abrupt increase of water depth (peak of transgression), and mark the time of maximum landward shift of the shoreline.

### 5.1.3. Interval 3 (ca. 4.5-0.3 ka cal. BP)





The recentmost interval (from 430 cm to the top of core) corresponds to seismic units U600 and U610 (Fig. 2), that formed

during the late Holocene a series of regressive deltaic lobes, that make up the Highstand Systems Tract in the sequence stratigraphic terminology. They consist in fine-grained prodeltaic deposits, and are related to the activity of the St Ferréol and Ulmet distributaries (seismic unit U600), and to the synchronous, then successive, activity of the Grand Passon and Bras de Fer Channels (seismic unit U610) (Fanget et al., 2014). At the core site, clay minerals are dominated by illite, as elsewhere in the Rhone prodelta (Chamley, 1971). Indeed, the Rhone River, receiving principally its detrital material from

the Alps, is particularly rich in illite, associated with some chlorite (Chamley, 1971) that tends to be trapped in sandy sediments during deposition (Chamley, 1971; Giresse et al., 2004). Both minerals represent the relative contribution of physical weathering to sedimentation, since they are resistant to degradation and transport (Chamley, 1971). Relative contents of illite are changing simultaneously with changes in sedimentary facies and activity of different distributaries (Fig. 5). Smectite contents are low as a whole but higher than within underlying intervals, when sea level was lower. The onset of

seaward progradation of the Rhone deltaic lobes corresponds, by definition, to the age of deposits situated immediately above the MFS. This age is ca. 4.5 ka cal. BP according to the age model. It corresponds to a marked increase of smectite content. It also corresponds to a marked increase in benthic foraminifera belonging to Cluster 2 (*C. carinata*, *B. marginata*, and *V. bradyana*) and ostracods belonging to cluster D (*C. rotundatum* and *Krithe* spp.) (Figs. 8 and 9). Foraminifera assemblage is constituted by typical species living in the distal part of the Rhone prodelta, with fine-grained sediments

enriched in both terrestrial and marine organic matter (Goineau et al., 2011; Goineau et al., 2015; Kruit, 1955; Mojtahid et al., 2009). They are also reported as opportunistic species able to respond quickly to fresh phytodetritus input by increased reproduction (De Rijk et al., 2000; Fontanier et al., 2003; Goineau et al., 2011; Jorissen, 1987). Ostracods content of Cluster D is known as common assemblage of circa-littoral to epi-bathyal environments (Bonaduce et al., 1975; Coles et al., 1994; Cronin et al., 1999; Didié et al., 2002; Yamaguchi and Norris, 2012). In the Rhone subaqueous delta, we hypothesize that

these species can be tolerant to moderate river influence (Fanget et al., 2013b). At the core site, strong decreases of Cluster 1 and Cluster B (shallow infra-littoral species), and increases of Cluster 2 and Cluster D reveal the establishment of prodeltaic conditions since 4.5 ka cal. BP (Figs 8 and 9). More precisely, they correspond to the progradation of the St Ferréol and Ulmet lobes. A similar pattern is identified on boreholes in the Rhone delta plain, where the onset of prodeltaic sedimentation is marked by the dominance of *V. bradyana* around 4 ka cal. BP (Amorosi et al., 2013).

Within Interval 3, we note also the presence of benthic foraminifera belonging to Cluster 3 (*H. depressula*, *A. beccarii*, *E. scabra*), and ostracods belonging to the genus *Leptocythere* and to Cluster E and F (Figs. 8 and 9). The vertical pattern of these ostracods in this interval will be discussed in further details in the next section (5.2). Foraminifera constituting Cluster 3 are typical of very shallow-water environments, and *E. scabra* is notably known to be adapted to thrive in organic matter-enriched and hypoxic sediments (Diz and Francés, 2008; Donnici and Serandrei Barbero, 2002; Jorissen, 1987; Mendes et

al., 2004). This assemblage increases in the uppermost 300 cm of the core, in concomitance with increased hydrotroilite content. Autigenic minerals generated by sulfate reduction (hydrotroilite) can be related both to high sedimentation rate (as observed in the core), leading to reducing conditions, and high organic matter input. These observations suggest increased



river influence that can be linked to the progressive progradation of Rhone delta, and to the beginning of activity of the Bras de Fer and Grand Passon Channels, located in front of the studied core.

## 5.2. Record of Holocene Cold Events (CRs)

Ostracods belonging to Cluster E and Cluster F, and especially to the genus *Leptocythere* show well-marked peaks within highstand prodeltaic deposits (Fig. 8). As previously described, the genus *Leptocythere* is widely distributed in brackish and shallow marine water environments (Anadon et al., 2002; Boomer and Eisenhauer, 2002; Carbonel, 1973, 1980; Frenzel and Boomer, 2005; Gliozzi et al., 2005; Van Morkhoven, 1963). In the Po delta, the occurrence of *Leptocythere* sp. is notably related to local increase of fluvial influence (Rossi, 2009), and in the Rhone delta, few valves of *Leptocythere* are encountered in restricted environmental areas characterized by estuarine conditions (Amorosi et al., 2013). Thus, the distribution pattern of *Leptocythere* through the highstand deposits would reflect hydrological fluctuations. During the Holocene, high fluctuations in precipitations are recorded, notably during the CRs (Mayewski et al., 2004; Wanner et al., 2014). In Europe, these CRs (or at least CR0, i.e. the 8.2 ka event, and CR6, i.e. the LIA) are characterized by intensified rainfalls (Arnaud et al., 2012; Magny et al., 2010; Magny and Begeot, 2004).

Two intervals of increased occurrence of *Leptocythere* (and therefore increased rainfall) are identified between ca. 400 and 350 cm, and ca. 70 and 0 cm (Fig. 8). They correspond to ages comprising between ca 4.0 and 2.2 ka cal BP and 0.6 and 0.2 ka cal. BP, respectively. These intervals are close to CR4 and CR6 (i.e. the LIA) that are dated between ca. 3.1 and 2.8 ka cal. BP and ca. 0.65 and 0.45 ka cal. BP, respectively (Wanner et al., 2014). The hypothesis of increased rainfall and river runoff, in the Rhone watershed, during the cooler late Holocene is supported, at least for the LIA, by observed advance of the Rhone Glacier (Goehring et al., 2011), high level of the Bourget Lake (France) (Arnaud et al., 2012), higher soil erosions in the French Pre-Alps (Simonneau et al., 2013), increased detritism in the Rhone delta plain (Bruneton et al., 2001; Provansal et al., 2003), and increased Rhone River floods (Pichard, 1995). In contrast, CR5, the so-called Migration Period Cooling, is not characterized by any increase in *Leptocythere*, suggesting dryer conditions in the Rhone watershed, compared to CR4 and CR6.

The signature of CR0 (the 8.2 ka event), CR1, CR2, and RCC3, is difficult to discriminate since they are incorporated within a condensed interval with very low accumulation rate (Figs 8).

On the other hand, we also notice that two other periods of increased *Leptocythere* are recorded between ca. 300 and 230 cm and between ca. 180 and 120 cm, *i.e.* between ca. 1.4 and 1.3 ka cal. BP and ca. 1.2 and 1.0 ka cal. BP (Fig. 8). These periods are not related to global events (CR-like), but might correlate to periods of regional climate deterioration as attested by high level of the Bourget lake (Arnaud et al., 2012), and by periods of increased detritism in the Rhone delta plain (Provansal et al., 2003).

Within the late Holocene interval, the distribution pattern of Cluster E is slightly offset of the single species *Leptocythere* (Fig. 8). This Cluster is constituted by the shallow infra-littoral *Paradoxostoma* and *Loxoconcha* species (Bonaduce et al., 1975; El Hmaidi et al., 2010), and by the epi-bathyal *Argilloecia* species. *Argilloecia* sp., in the Rhone subaqueous delta,





appears to be tolerant to fluvial influence and respond potentially to organic matter supply (Fanget et al., 2013b). Nevertheless, increased of Cluster E is not recorded within the periods characterized by higher river supply such as the CR4 and CR6 (i.e. the LIA), but slightly after these periods. It possibly indicates that *Leptocythere* is a better competitor during periods of increased detritism and fluvial discharge.

**6. Conclusion**

Our study shows that some environmental and sea-level changes during the Holocene can be clearly depicted from sedimentological and benthic meiofauna proxies.

During the early Holocene (11.7 to 7-8 ka cal. BP), sea-level rise is led to the deposition of tempestite sediments that contain shallow infra-littoral benthic meiofauna. These deposits are thought to be formed between ca. 20 and 50 m water depth, and we believe that this feature can be used as a good regional scale paleobathymetric marker.

The middle Holocene (7-8 to 4-5 ka cal. BP) corresponds to a phase of very low sedimentation at the core site, resulting in the formation of a condensed interval (i.e. the Maximum Flooding Surface in a sequence stratigraphic terminology) reflecting the further landward position of the shoreline and Rhone outlet(s). This MFS contains reworked shoreface material within a fine-grained matrix. It displays mixed faunal assemblages, ranging from infra-littoral to epi-bathyal environments, which are the result of erosion processes that occurred during the period of transgressive submersion and, then, mark the peak of transgression and the subsequent sediment starvation.

Following the transgressive maximum, the late Holocene (4-5 ka cal. BP to 19[th] century AD) sediment deposits are influenced by a combination of allocyclic and autocyclic factors. The progressive shoreline progradation and prodeltaic lobes switching are characterized by the setting up of benthic meiofauna adapted to thrive in the distal part of the Rhone River influence (i.e. distal St Ferréol and Ulmet lobes), and by the presence of very shallow-water species (i.e. proximal Grand Passon and Bras de Fer lobes).

Within the late Holocene deposits, ostracod assemblages emphasize fluctuations in the Rhone River hydrological activity. In particular, the occurrence of the ostracod genus *Leptocythere* highlights periods of increased fluvial discharge. These periods of intensified runoffs can be attributed to the 2.8 ka event (CR4) and the Little Ice Age (CR6) that are known to be at the origin of regional climate deterioration in Western Europe, as well as periods of regional climate deterioration at ca. 1.3 and 1.1 ka cal. BP. In contrast, the signature of the early and middle Holocene cold relapses are difficult to explore in the Rhone subaqueous delta since they correspond respectively to (a) a phase of rapid sea-level rise at the origin of shoreline reworking and deposition of tempestite, and (b) a period of very low sedimentation at the core site resulting in a condensed interval with low temporal resolution.

Finally, our study demonstrates that prodeltas may provide interesting expanded archives of climate changes at the land/sea interface, with accumulation rates reaching 0.4 m yr[-1]. On the other hand, such resolution can be achieved at one single site for only short time-intervals, since depot-centers migrate rapidly in response to sea-level changes, high sediment fluxes and



lateral shifting of deltas lobes. This highlights the need of acquiring series of long cores/boreholes, parallel and orthogonal to deltaic systems.

## Acknowledgments

Core RHS-KS55 was collected during the RHOSOS cruise (2008) on board R/V *Le Suroît*. We thank the captain and crew of this cruise together with the Genavir technical staff as well as the scientific parties. Special thanks are due to Bernard Dennielou (IFREMER, Brest), for his commitment at sea and during the processing of the data in the laboratory. We thank the *Laboratoire de Mesure du Carbone 14*, UMS 2572, ARTEMIS in Saclay for 14C measurements by SMA in the frame of the National Service to CEA, CNRS, IRD, IRSN and *Ministère de la Culture et de la Communication*. This work was partly supported by the CNRS-INSU "Mistrals-Paleomex". We are grateful to Bertil Hebert (CEFREM, University of Perpignan), and Gilbert Floch, Angélique Roubi and Mickaël Rovere (IFREMER, Brest) for their technical support.

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



| Event | Time slice (kyr) | References |
|:---:|:---:|:---:|
| CR0 | 8.2 | Barber et al. (1999) |
| CR1 | 6.4-6.2 | Wanner et al. (2011) |
| CR2 | 5.3-5.0 | Magny and Haas (2004), Roberts et al. (2011) |
| CR3 | 4.2-3.9 | Walker et al. (2012) |
| CR4 | 2.8-3.1 | Chambers et al. (2007), Swindles et al. (2007) |
| CR5 | 1.45-1.65 | Wanner et al. (2011) |
| CR6 | 0.65-0.45 | Wanner et al. (2011) |

**Table 1: Chronology of Holocene cold relapses (CR) based on existing literature.**





| Depth (cm) | Material | Weight (mg) | Sample number | 14C conventional age (yr BP) | Calibrated age (yr cal. BP) | Mean calibrated age (yr cal. BP) |
|---|---|---|---|---|---|---|
| 90-93 | Benthic foraminifera | 9.5 | SacA 27201 | 1335 ± 30 | 790-944 | 867 |
| 120-123 | Benthic foraminifera | 9.9 | SacA 27202 | 1840 ± 30 | 1300-1477 | 1389 |
| 150-153 | Benthic foraminifera | 11 | SacA 23204 | 2080 ± 35 | 1551-1761 | 1656 |
| 200-203 | Benthic foraminifera | 10.6 | SacA 23205 | 1655 ± 30 | 1154-1282 | 1218 |
| 300-303 | Benthic Foraminifera+ *Turritella* sp | 10.9 | SacA 23206 | 1900 ± 30 | 1362-1527 | 1445 |
| 335-336 | Benthic Foraminifera+ *Turritella* sp+ mixed bivalves | 10 | SacA 23207 | 1705 ± 30 | 1185-1318 | 1252 |
| 350-353 | Benthic foraminifera | 11.3 | SacA 27203 | 2760 ± 35 | 2351-2619 | 2485 |
| 417-420 | *Turritella* sp | 896 | Poz-35061 | 4335 ± 35 | 4375-4574 | 4475 |
| 430-433 | Benthic foraminifera | 10.5 | SacA 27204 | 6190 ± 40 | 6513-6735 | 6624 |
| 440-443 | *Nucula* sp | 11.2 | SacA 27205 | 7830 ± 40 | 8192-8374 | 8283 |
| 470-473 | Benthic foraminifera | 10.2 | SacA 23208 | 8565 ± 35 | 9075-9333 | 9204 |
| 510-513 | *Elphidium crispum* | 10.1 | SacA 23209 | 10790 ± 40 | 12058-12467 | 12263 |
| 670-673 | Benthic foraminifera | 13 | SacA 23210 | 11855 ± 45 | 13215-13433 | 13324 |
| 730-733 | *Elphidium crispum* | 10.6 | SacA 23211 | 11280 ± 40 | 12644-12870 | 12757 |

**Table 2: Summary of 14C dates. Absolute dates were obtained with accelerator mass spectrometer (AMS) 14C dating of well-preserved shells and benthic foraminifera at Laboratoire de Mesure 14C (LMC14) at Commissariat à l'Energie Atomique (CEA, Saclay) and at Poznan Radiocarbon Laboratory (PRL). The ages reported herein are delta 13C-normalised conventional 14C years, corrected for an assumed airsea reservoir effect of 400 years. Calendar ages were calculated using clam software (Blaauw, 2010) and the Marine 13 calibration curve (Reimer et al., 2013).**




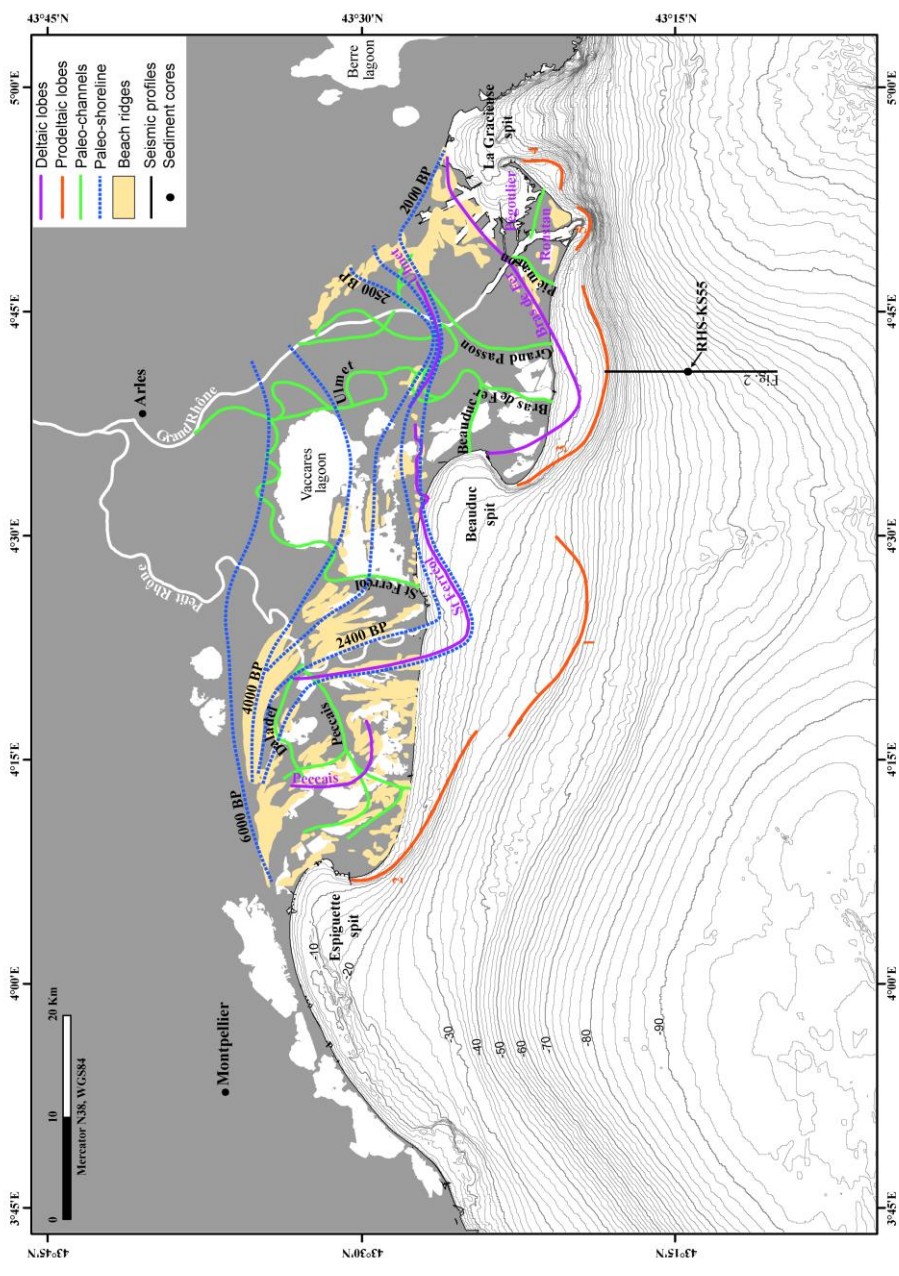

**Figure 1: Offshore and onshore morphology of the Rhone deltaic system. This map is based on the compilation of Berné et al. (2007) and Vella et al. (2008). The successive shifting of the Rhone distributaries under natural and/or anthropic influence during the middle and late Holocene led to the formation of several deltaic lobes. Steps in the present-day morphology correspond to the position of the paleo-delta fronts that can be linked to known paleo-distributaries of the Rhone: (1) early Saint Ferréol; (2) Peccais; (3) Bras de Fer; (4) Pégoulier (5) and modern Roustan distributary. The map of relict morpho-sedimentary units in the Rhone delta plain (paleo-shorelines, beach ridges and onshore deltaic lobes) is based on L'Homer et al. (1981), Arnaud-Fassetta (1998), Vella (1999) and Provansal et al. (2003). Thick line and black dot correspond respectively to chirp seismic profile (Fig. 2) and sediment core RHKS-55 presented in this study.**





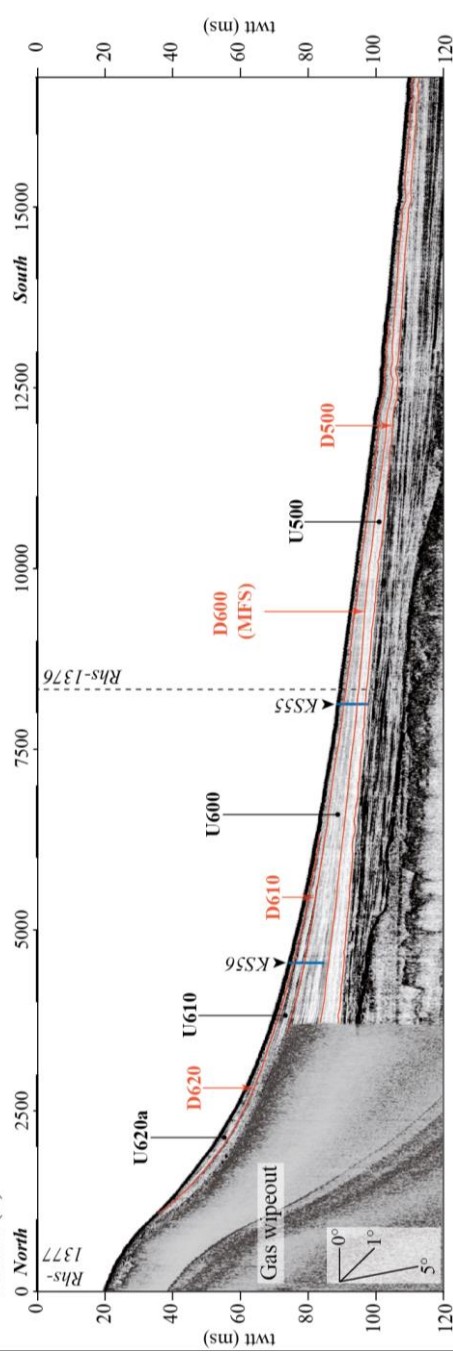

**Figure 2: "Dip" (NS) chirp seismic profile across the Grand Passon and Bras de Fer subaqueous delta (position in Fig. 1). Surface D500 corresponds to the flooding surface (that coincides here with a wave ravinement surface) that separates the Younger Dryas (seismic unit U400) deposits from the Preboreal deposits (seismic unit U500). The downlap surface D600 is the Maximum Flooding Surface formed during the turnaround between retrogradation and progradation. D610 and D620 are erosional surfaces that mimic flooding surfaces and separate the middle and late Holocene sedimentary wedges (seismic units U600, U610 and U620a) formed in response to the successive shifts of Rhone Channel.**




**Figure 3: Age model for the upper part (middle and late Holocene) of core RHS-KS55 based on linear interpolation using the**
**Clam 2.2 software (Blaauw, 2010).**



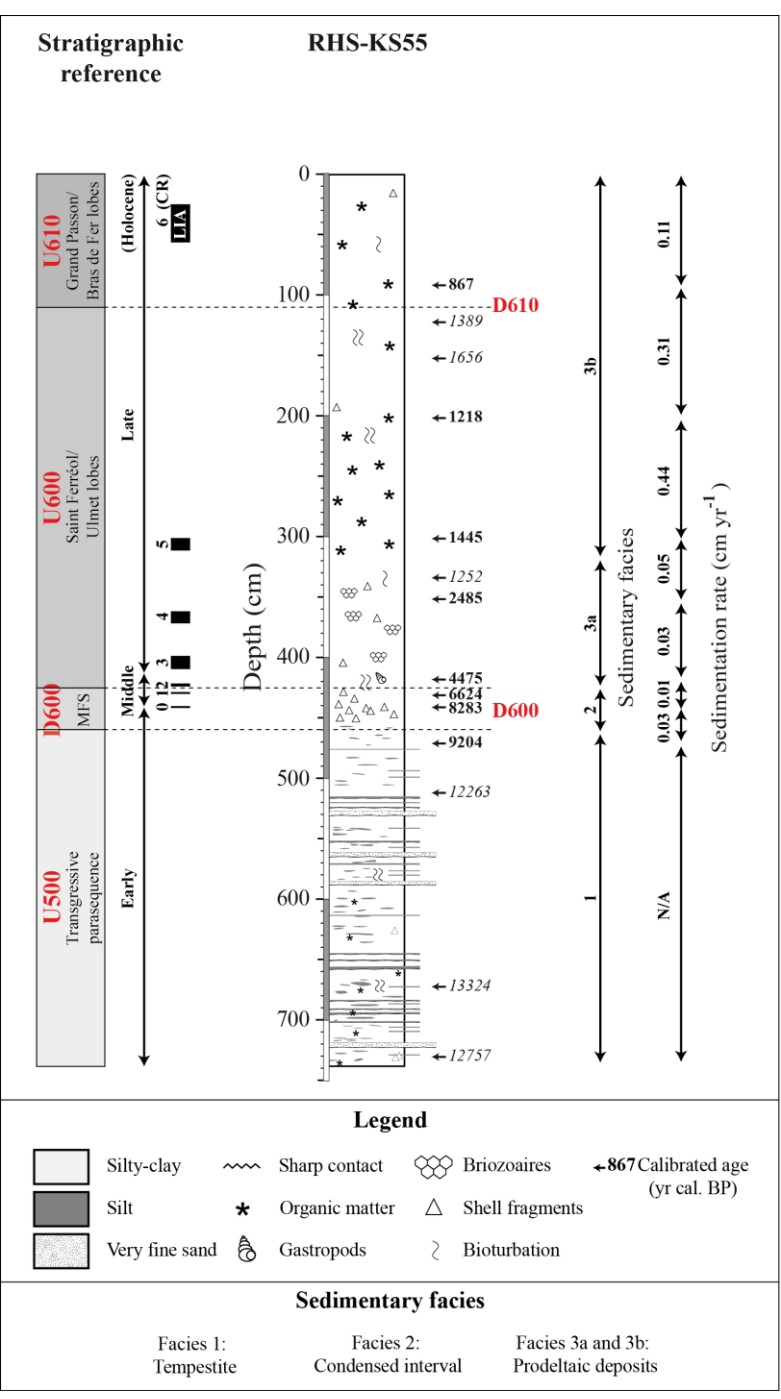

**Figure 4: Sediment features and sedimentation rates of core RHS-KS55. Correlations with seismic units and the Holocene chronology are shown. Black rectangles on the left side of the figure represent the most significant periods of climate deterioration (known as Cold Relapses (CR), Wanner et al., 2014) during the Holocene (e.g. the 8.2 ka event, the Little Ice Age).**




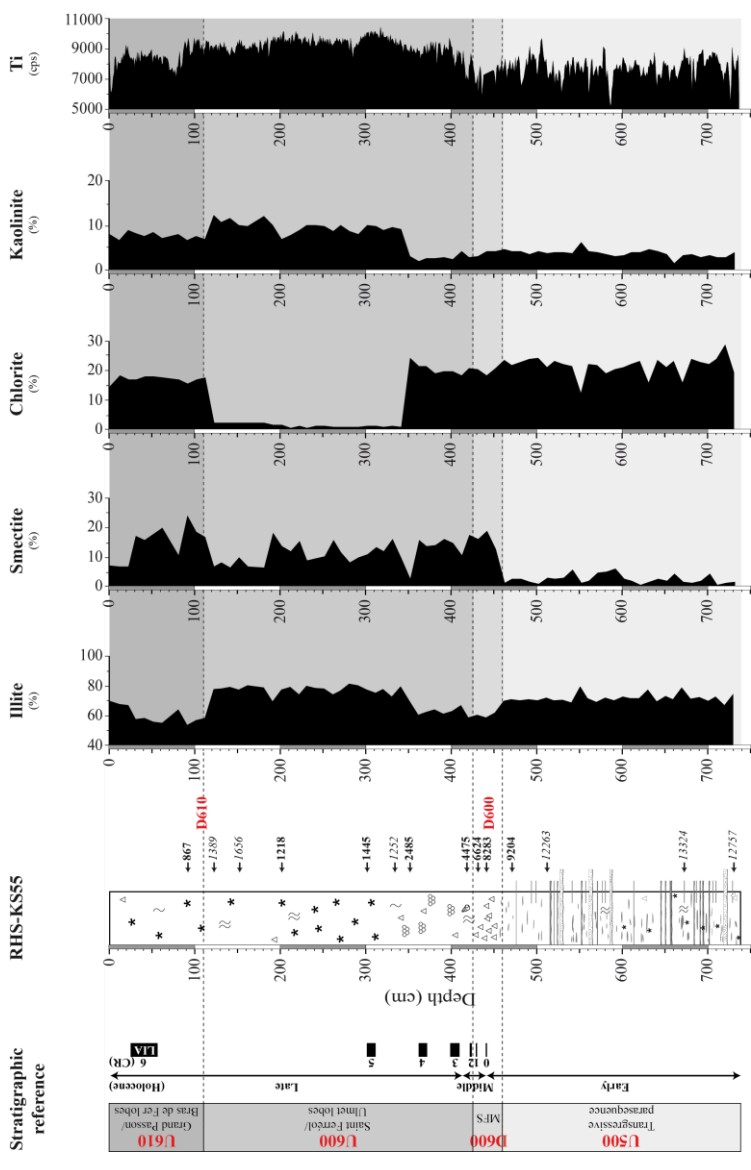


**Figure 5: Distribution of the main clay minerals and bulk intensity of Ti in core RHS-KS55. Correlations with seismic units and Holocene cooling events are shown on the left side of the figure.**





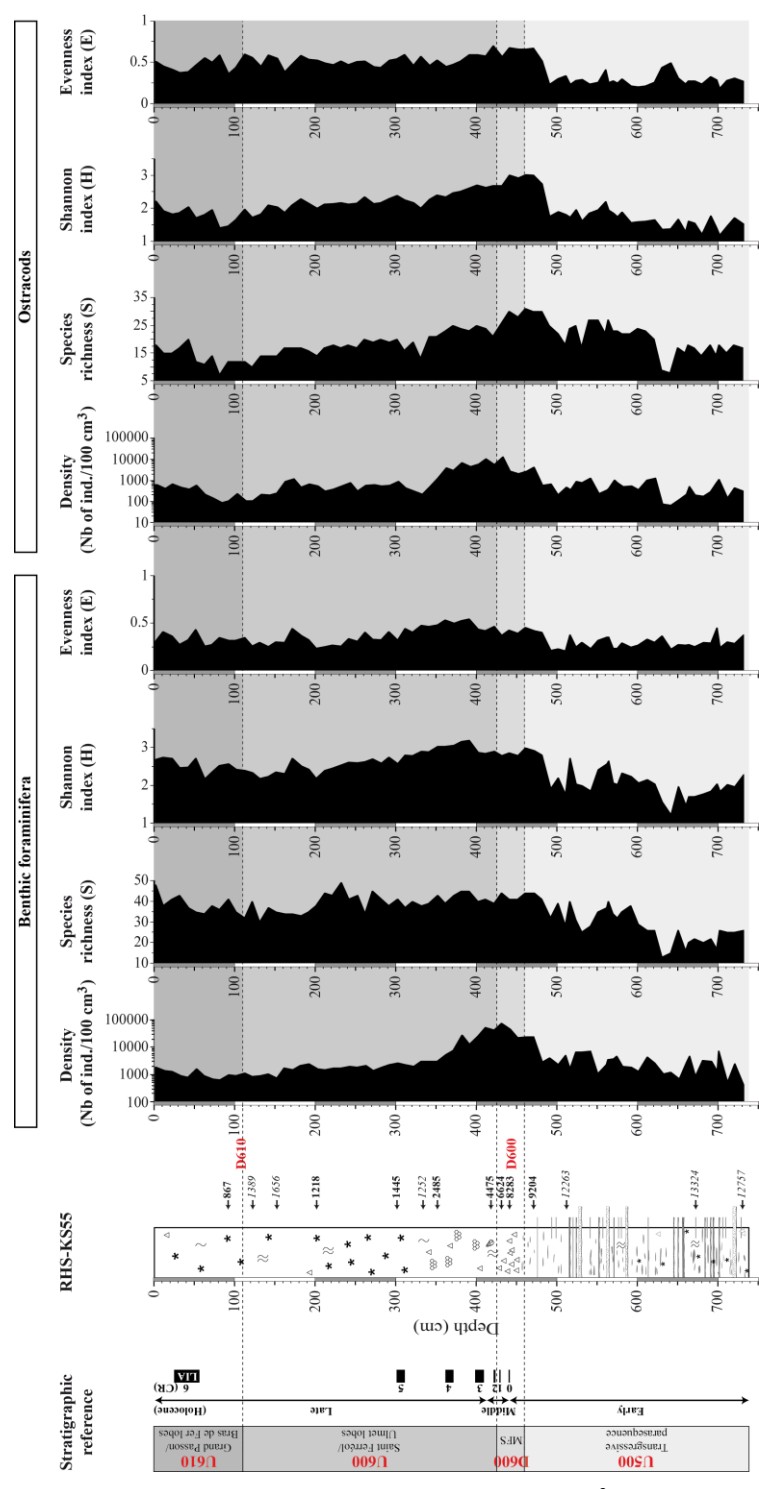

**Figure 6:** Ecological indices (total abundance per 100 cm$^3$, number of species (S), Shannon (H) and Evenness (E) index) describing benthic foraminiferal and ostracod populations. Correlations with seismic units and Holocene cooling events are shown on the left side of the figure.



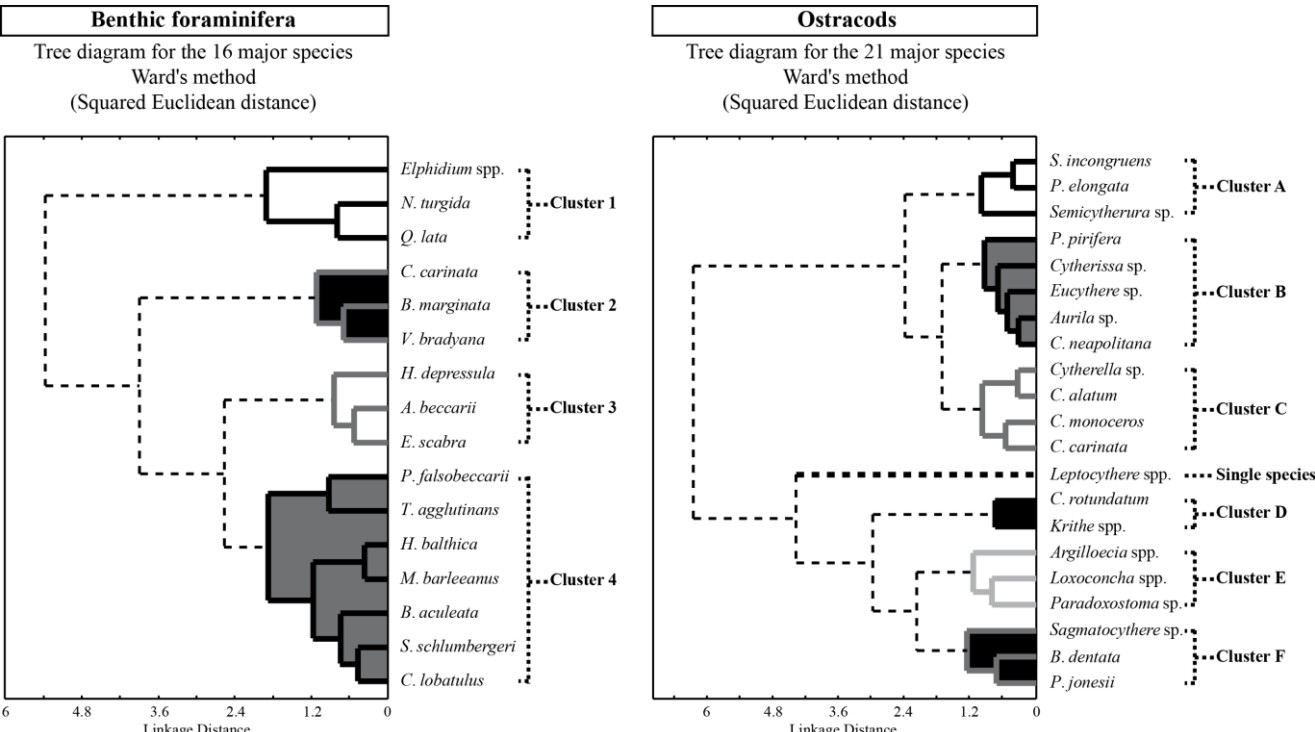

**Figure 7: R-mode cluster analyses of the 16 major (more than 5% in at least one sample) benthic foraminiferal species, and of the 21 major ostracod species according to Ward's method, based on standardized percentages pi of these species.**

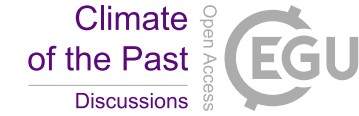



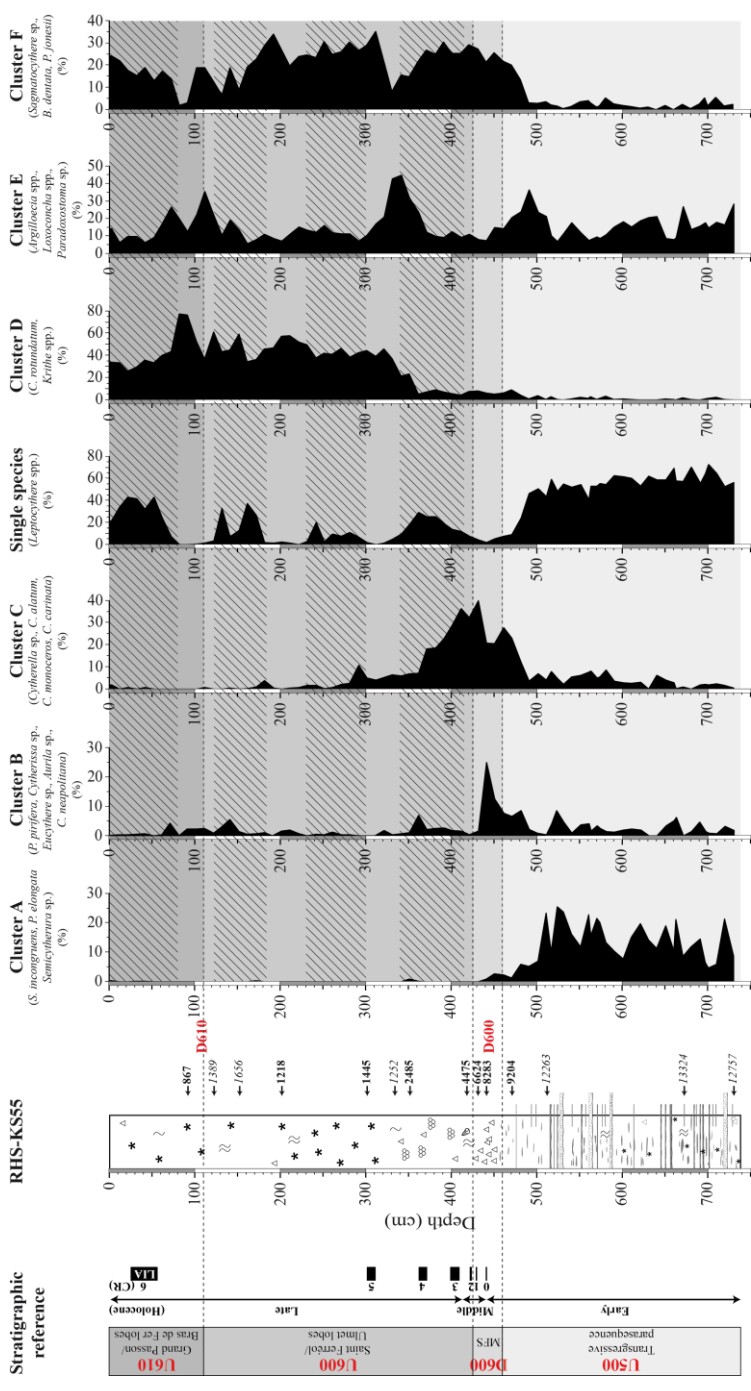

**Figure 8: Cumulative percentages of taxa composing the defined ostracod clusters along core RHSKS55. Correlations with seismic units and Holocene cooling events are shown on the left side of the figure. Dashed lines highlight periods of increased fluvial discharge at the core site.**




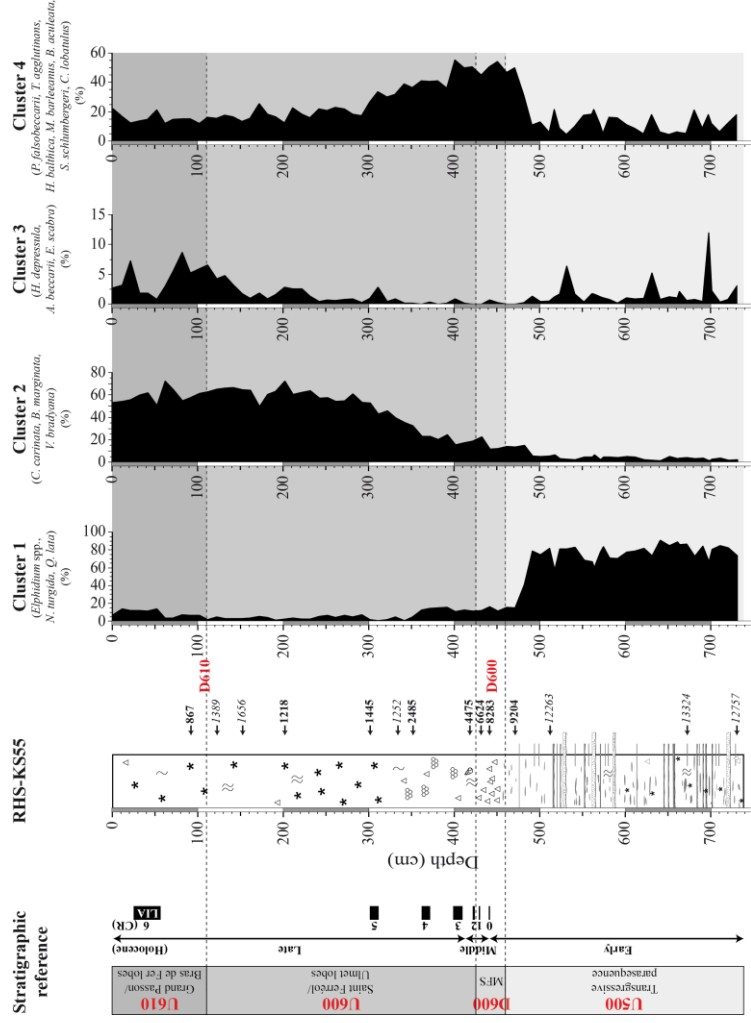

**Figure 9: Cumulative percentages of taxa composing the defined benthic foraminiferal clusters along core RHS-KS55. Correlations with seismic units and Holocene cooling events are shown on the left side of the figure.**