# Peer review of "Sedimentary archives of climate and sea-level changes during the Holocene in the Rhone prodelta (NW Mediterranean Sea)"

_Climate of the Past, 2016_

## Short Comment (SC1) · 10 Jul 2016

I enjoyed studying this very interesting study. I am particularly interested in the climate of the past 1000 years and noticed the dry phase that is documented in the dataset for the Medieval Climate Anomaly (MCA) by the low abuncance of Leptocythere which serves as proxy for increased fluvial discharge. The MCA dry phase at this location fits well with other studies from the Mediterranean. See yellow dots here: http://t1p.de/mwp It might be worth adding the MCA observation to the text, as the Little Ice Age is already mentioned as wet phase.

Figure 8 is a key figure and needs a better and more detailed figure caption. Increased fluvial discharge (Leptocythere peaks) are marked by 'hatched' (not 'dashed') patterns.

[Figure]

Did I understand this correctly?

In general it would be great if a summary figure could be shown where the key climatic findings are shown with the y-axis being time in years BP.

———————————————

---

## Referee Comment (RC1) · A. Amorosi (Referee) · 29 Jul 2016

**General comments**

This paper provides convincing evidence of multidecadal to century-scale record of sea-level and climate change from the Holocene succession of the Rhône prodelta, in southern France. Based on a multiproxy record from a 7 m-long core, the authors provide nice documentation of transgressive and highstand sedimentation, focusing on a condensed interval that spans a 4 kyr interval of time. This study clearly documents how subtle changes in the meiofauna (benthic foraminifers and ostracods) can be used as a proxy for both sea-level and climate change. In particular, though I'm not a fossil specialist, I enjoyed the way ostracods were used on very high-resolution time scales

as indicators of hydrological fluctuations.

Fossil analysis is very accurate, and in general, I found the data presented and the inferences drawn to be convincing. In my opinion, this paper can be published following minor revisions. A few points that might deserve further attention are highlighted below

Specific comments

The authors emphasize the multi-proxy character of their study, which includes sedimentology, paleontology, mineralogy and geochemistry. Mineralogical interpretation, however, is rather vague and geochemical analysis is restricted to a single element (Ti) profile, with no discussion/interpretation in text. The sedimentological and paleontological aspects of this paper are very robust and fully support the conclusions. Changes in sediment composition are very important, but reliable interpretations probably require a significantly larger data set (to the scale of the source-to-sink system) than the one available for this study. I think this part could be removed without detriment.

In terms of sequence stratigraphy, what the authors call the "maximum flooding surface" (MFS) should be termed more properly (at the core scale) the "condensed section" (CS). CS is the label that sequence stratigraphy uses to describe exactly what the authors see in their cores: a few dm-thick, condensed stratigraphic interval that marks the inversion from a transgressive to a "regressive" depositional trend. On a seismic scale, the authors can use the term "MFS" for this very thin stratigraphic interval. On the very high-resolution scale of their study, however, I feel that the term "CS" would be more appropriate (see comment below).

I can't see "very distinct changes in all proxy records at the maximum flooding surface". There is clearly an inversion that can be seen across the condensed section along most profiles, but these changes seem to occur gradually. Several mineralogical proxies show remarkable changes a few dm above the condensed section, and not at its top. Similarly, changes along fossil profiles occur even below the CS, and not necessarily at its base. Given the definition of the maximum flooding surface, which simply marks

the turnaround from transgression to regression, there is no reason to record sharp changes across this surface, while opposite trends are clearly expected above and below the MFS/CS. In this regard, the data shown by the various profiles are fully consistent with the authors interpretation.

Last point: the authors reject three "old" radiometric dates from the lower part of their core (a > 2m-thick stratigraphic interval), based on regional correlations of unit U500. For convincing the reader of their interpretation, they could probably expand slightly on this, stating how many radiocarbon dates from unit U500 are regionally available and, especially, which sedimentary facies did they find in other cores penetrating the same unit (was it the same deposits or something different?)

---

## Referee Comment (RC2) · L. Giosan (Referee) · 1 Aug 2016

This is an excellent paper all around that shows the potential of distal delta deposits for paleoenvironmental reconstructions. Here are some weaker points that need to be addressed:

1. the paper needs a better justification for radiocarbin data rejection.

2. inversions in radiocarbon ages are indicative of reworking, which is a well known fact of life in these environments regardless of the facies discussed in the paper. Note that I am not talking about reworking of very old microfosssils that show on their morphology or color that they are old and reworked; I am talking of specimens that could be 1000-

2000 years older and look like new. But 1000 years is a long time in the Holocene. In these conditions the paper needs a discussion on reworking and transport of microfossils used in this study. Do they matter and how much? Can reworked shallow species mimic a hydrological event? This request might seem hard but it is important if prodeltaic records are to be used for paleohydrology. And the authors have the right data to make a good case that reworking is secondary.

I am looking forward to read the revised discussion inclduing these points.

Liviu Giosan Geoscientist Woods Hole Oceanographic Institution

---

## Editor Comment (EC1) · N. Combourieu Nebout (Editor) · 2 Sep 2016

Dear authors,

We have now received two reviews of your paper and a free comment. All two reviewers make comments you have to follow.

You have now to post your replies on the discussion forum and respond to all comments explaining how you want to modify your manuscript if necessary. Please also prepare a revised version of your paper accordingly. In the revised version, I would like to see your corrections in track change mode.

sincerely Yours

[Figure]

Nathalie Combourieu-Nebout

---

## Author Comment (AC1) · 4 Oct 2016

**Sebastian Luening - Interactive comment**

I enjoyed studying this very interesting study. I am particularly interested in the climate of the past 1000 years and noticed the dry phase that is documented in the dataset for the Medieval Climate Anomaly (MCA) by the low abundance of Leptocythere which serves as proxy for increased fluvial discharge. The MCA dry phase at this location fits well with other studies from the Mediterranean. See yellow dots here: http://t1p.de/mwp. It might be worth adding the MCA observation to the text, as the Little Ice Age is already mentioned as wet phase. Figure 8 is a key figure and needs a better and more detailed figure caption. Increased fluvial discharge (Leptocythere peaks) are marked by 'hatched' (not 'dashed') patterns.

Did I understand this correctly? In general it would be great if a summary figure could be shown where the key climatic findings are shown with the y-axis being time in years BP.

Thank you for your comment on the Medieval Climate Anomaly. We added a paragraph in the discussion part of our paper describing the signature of the MCA at our studied site.

*"Conversely, a strong decrease in Leptocythere is observed from 120 to 80 cm (Fig. 8). It suggests dryer conditions during this interval which corresponds to the Medieval Climate Anomaly (ca. 950-1250 AD). In the Northern Hemisphere, the MCA is generally described as a warm period characterized by intense dryness. In the Mediterranean, several studies highlighted dryer conditions during this event (e.g. Wassenburg et al., 2013; Martinez-Ruiz et al., 2015; Bassetti et al., 2016). The same signature is also observed in the Alps and the Rhone watershed, with periods of low lake level and low flood frequency, respectively (e.g. Magny, 2004; Wilhelm et al., 2016). Thus, the hypothesis of increased drought at the studied site during the MCA fits well with regional and local observation."*

For better understanding, we also detailed the figure caption of figure 8 and changed the term "dashed" to "hatched"

*"Figure 8: Cumulative percentages of taxa composing the defined ostracod clusters along core RHSKS55. Correlations with seismic units are shown on the left side of the figure. Holocene Cold Relapses are also indicated on the y-axis. Periods of increased fluvial discharge are highlighted by peaks in Leptocythere. These periods of intensified runoffs correlate to the 2.8 ka event (CR4) and the LIA (CR6). Two periods of regional climate deterioration are also observed at 1.3 and 1.1 ka cal. BP. Conversely, the Migration Period*

*Cooling (CR5) and the Medieval Climate Anomaly (MCA) corresponds to period of increased dryness."*

---

## Author Comment (AC2) · 4 Oct 2016

**Alessandro Amorosi - Interactive comment**

**General comments**

This paper provides convincing evidence of multidecadal to century-scale record of sea-level and climate change from the Holocene succession of the Rhône prodelta, in southern France. Based on a multiproxy record from a 7 m-long core, the authors provide nice documentation of transgressive and highstand sedimentation, focusing on a condensed interval that spans a 4 kyr interval of time. This study clearly documents how subtle changes in the meiofauna (benthic foraminifers and ostracods) can be used as a proxy for both sea-level and climate change. In particular, though I'm not a fossil specialist, I enjoyed the way ostracods were used on very high-resolution time scales as indicators of hydrological fluctuations.

Fossil analysis is very accurate, and in general, I found the data presented and the inferences drawn to be convincing. In my opinion, this paper can be published following minor revisions. A few points that might deserve further attention are highlighted below.

**Specific comments**

The authors emphasize the multi-proxy character of their study, which includes sedimentology, paleontology, mineralogy and geochemistry. Mineralogical interpretation, however, is rather vague and geochemical analysis is restricted to a single element (Ti) profile, with no discussion/interpretation in text. The sedimentological and paleontological aspects of this paper are very robust and fully support the conclusions. Changes in sediment composition are very important, but reliable interpretations probably require a significantly larger data set (to the scale of the source-to-sink system) than the one available for this study. I think this part could be removed without detriment.

We agree with Alessandro Amorosi that geochemical analysis in the present study was too vague (restricted to Titanium element) and did not really support our results and discussion. Thus, we decided to remove XRF data from our paper.

Even if clay mineral assemblages appeared to be less robust than micropaleontological data, we would prefer to not remove this aspect of our paper. Indeed, we observe that changes in clay mineralogy through the core are closely related to the successive migrations of the Rhone River. It also appeared that the condensed section is characterized by a distinct change in clay mineralogy with a decrease in illite content and an increase in smectite content. Nevertheless, we agree that a larger dataset would be useful in the future to discuss in detail changes in

sediments composition and allow a source to sink record and understanding of sediment composition evolution through time. But only few clay mineral studies have been carried out in the Rhone watershed and Gulf of Lions in the past. This dataset is thus considered as important even if detail interpretation of this one is yet difficult.

In terms of sequence stratigraphy, what the authors call the "maximum flooding surface" (MFS) should be termed more properly (at the core scale) the "condensed section"(CS). CS is the label that sequence stratigraphy uses to describe exactly what the authors see in their cores: a few dm-thick, condensed stratigraphic interval that marks the inversion from a transgressive to a "regressive" depositional trend. On a seismic scale, the authors can use the term "MFS" for this very thin stratigraphic interval. On the very high-resolution scale of their study, however, I feel that the term "CS" would be more appropriate (see comment below).

We modified the term "Maximum Flooding Surface" in the text. We used only the term MFS when we discussed seismic data. When we are discussing sedimentological data we changed it to "condensed section" as suggested.

I can't see "very distinct changes in all proxy records at the maximum flooding surface". There is clearly an inversion that can be seen across the condensed section along most profiles, but these changes seem to occur gradually. Several mineralogical proxies show remarkable changes a few dm above the condensed section, and not at its top. Similarly, changes along fossil profiles occur even below the CS, and not necessarily at its base. Given the definition of the maximum flooding surface, which simply marks the turnaround from transgression to regression, there is no reason to record sharp changes across this surface, while opposite trends are clearly expected above and below the MFS/CS. In this regard, the data shown by the various profiles are fully consistent with the authors interpretation.

We agree that not all proxies show a sharp change at the transition between Unit U500 and the condensed section. Except for the clay mineral assemblages which present a very distinct change (decrease in illite content and increase in smectite content) at this transition, micropaleontological proxies are rather characterized by a gradual change through this transition. Therefore we modified this sentence.
*"The transition between the early Holocene deposits and the middle Holocene condensed section is marked by a gradual change in all proxy records."*

Last point: the authors reject three "old" radiometric dates from the lower part of their core (a > 2m-thick stratigraphic interval), based on regional correlations of unit U500. For convincing the reader of their interpretation, they could probably expand slightly on this, stating how many radiocarbon dates from unit U500 are regionally available and, especially, which sedimentary facies did they find in other cores penetrating the same unit (was it the same deposits or something different?).

Indeed, we rejected three "old" 14C dates from the lower part of the studied core. We consider that these 14C dates are too old for seismic unit U500. Considering the age of the underlying deposits of seismic unit U400 in this area (ca. 10.5 ka cal. BP; Berné et al., 2007), we can only assume that 14C dates obtained in unit U500 are generally biased. At the regional scale, this transgressive parasequence has been described in all the studied cores as a unit made of tempestite deposits (Fanget et al., 2014). Thus, it consists of numerous silt or very fine sand laminae interlaminated with silty clay. Within these deposits, benthic foraminifera (picked for 14C dates) were mainly represented by infra-littoral (upper shoreface) species (*Elphidium crispum*). These benthic foraminifera are likely reworked by high energy hydrodynamic processes and incorporated into modern deposits. As a consequence, these 14C dates are too old. Only 14C dates from the upperpart of unit U500 (i.e. five 14C dates at the regional scale, Fanget et al., 2014) are considered as reliable. It is explained by a gradual change in lithology (increase in clay content and decrease in silt and fine sand laminae) within the upperpart of unit U500, which reflects a deepening of depositional setting and a reducing of reworking processes.

We added some information in the text to explain in more details why we rejected these 14C dates. We also added a paragraph in the discussion section (5.1.) to discuss in more details reworking processes in subaqueous deltaic environment.

*"Considering the age of the underlying deposits of seismic unit U400 in this area (i.e. paleo-deltaic complex of the Rhone (ERDC), ca. 10.5 ka cal. BP, Berné et al., 2007), we assume that 14C dates obtained in this unit are generally biased, because of reworking occurring in shallow water environments. This interpretation is supported by the nature of sediments that composed this unit. Indeed, at the regional scale, seismic unit U500 is made of tempestite deposits (Facies 1, section 4.2.) which mainly contain infra-littoral (i.e. upper shoreface) benthic foraminifera (Elphidium crispum; see Table 2 and Fanget et al., 2014). These benthic foraminifera are likely reworked by high energy hydrodynamic processes. The older ages*

*obtained within seismic unit U500 are thus probably the result of reworking during the transgression of an underlying Younger Dryas/Preboreal delta front (for more details, see section 5.1)"*

---

## Author Comment (AC3) · 4 Oct 2016

**Liviu Giosan - Interactive comment**

This is an excellent paper all around that shows the potential of distal delta deposits for paleoenvironmental reconstructions. Here are some weaker points that need to be addressed:

1. the paper needs a better justification for radiocarbon data rejection.

2. inversions in radiocarbon ages are indicative of reworking, which is a well-known fact of life in these environments regardless of the facies discussed in the paper. Note that I am not talking about reworking of very old microfosssils that show on their morphology or color that they are old and reworked; I am talking of specimens that could be 1000- 2000 years older and look like new. But 1000 years is a long time in the Holocene. In these conditions the paper needs a discussion on reworking and transport of microfossils used in this study. Do they matter and how much? Can reworked shallow species mimic a hydrological event? This request might seem hard but it is important if prodeltaic records are to be used for paleohydrology. And the authors have the right data to make a good case that reworking is secondary.

Within high energy transgressive deposits, reworking processes appear important, and will directly affect AMS dating in incorporating reworked benthic meiofauna into modern benthic meiofauna assemblages. Within highstand deposits, reworking processes are also regularly observed in shallow-water environments. They are thought to be the result of transport processes during periods of increased river discharge. They are thus transported further offshore within the river plume. The distribution pattern of these reworked benthic meiofauna within highstand deposits can directly reflect hydrological fluctuations in the past.

We added a paragraph at the beginning of the discussion to discuss reworking and transport of microfossils in shallow-water environments.

*"In subaqueous deltaic environments, reworking processes appear to be common within transgressive deposits (Cattaneo and Steel, 2003). In the Rhone subaqueous delta, transgressive deposits consist of tempestite deposits (seismic unit U500) which are the result of regular occurrence of high energy hydrodynamic processes (including combined storm and flood events; Fanget et al., 2014). These processes regularly winnowed the seafloor and generate erosion, reworking and transport of sediments. Thus, it is likely that benthic calcareous meiofauna are reworked from older deposits into modern deposits having the same faunal assemblages (Cearreta and Murray, 2000). These reworked benthic meiofauna cannot be considered as in situ, but it appears impossible to distinct them from the*

*unreworked modern tests and carapaces. It will directly affect AMS dating with measured ages older than true ages, as observed within the transgressive seismic unit U500. Such phenomena have been observed in Denmark (Heier-Nielsen et al., 1995) and in Spain (Cearreta and Murray, 2000), and highlight the difficulty to obtain reliable AMS dates from high energy transgressive deposits.*

*Within the recent most prograding units of the Highstand Systems Tract (4.5 to 0.3 ka cal. BP in the present study), we also observe the regular occurrence of reworking and transport of benthic meiofauna. Reworking processes are regularly encountered in shallow-water environments (e.g. Frenzel and Boomer, 2005; Loureiro et al., 2009; Fanget et al., 2013a). Conversely to the Transgressive Systems Tract, reworked benthic meiofauna are easier to identify since they originate from shallow-water environments and deposit into deeper settings. Reworking processes in AMS dating are thus considered as less important and problematic in Highstand Systems Tract. It is likely than these allochthonous benthic meiofauna are transported and redeposited further offshore within the river plume during periods of increased river discharge (Fanget et al., 2013a). Thus, it can be relevant to use allochthonous meiofauna as bio-markers for better understanding transport and reworking processes (Cronin, 1983; van Harten, 1986; Zhou and Zhao, 1999; Fanget et al., 2013a; Angue Minto'o et al., 2015), and study paleo-hydrology. The distribution pattern of reworked benthic meiofauna through highstand deposits is likely to reflect hydrological fluctuations in the past (see section 5.3.)."*

I am looking forward to read the revised discussion including these points.

---

## Author Comment (AC4) · 4 Oct 2016

Dear Editor,

Please find the revised version of our manuscript cp-2016-57. The manuscript has been revised taking into account the reviewers comments. You will find all the corrections in the "track changes" manuscript. I look forward to hearing from you and if you have any further queries please do not hesitate to contact me.

Yours sincerely,

Anne-Sophie Fanget

Please also note the supplement to this comment:
http://www.clim-past-discuss.net/cp-2016-57/cp-2016-57-AC4-supplement.pdf

[Figure]

**Supplement:**

[revised manuscript text omitted]

---

## Author Response (AR1)

Dear Nathalie Combourieu-Nebout,

Thank you for your comments. We changed the manuscript and figures according to your remarks.

1 - Regarding to the comment about the dates of the base of the series, I would like that you explain more. I understand that these old dates have been removed because they may be related to reworked material. Nevertheless, first I would like to see the number of ages removed clearly noted in the text (I think it is three?) and the mention of their values. I think that the fig 4 has to be linked to this explanation paragraph to show to the reader where theese removed ages are. This implies for you to have a look on the citation of figure 3 and 4 as they will be probably inverted. In addition could you please indicate in the table 2 the removed ages by a star or anything else? Secondly could you justify more than with one sentence why you consider that all three ages are wrong. Two of them have been performed on Elphidium and the third (670-673 cm) on benthic foraminifers without any species mentioned. I think that your explanation about the reworked processes have to be already developed here and not only in the discussion and the reader will better understand why you have rejected these dates in this part of the manuscript.

We gave more details about reworked $^{14}$C dates in the section Material and Methods and in the Section 4.1. We hope it would help to clarify our explanations. We indicated the number of $^{14}$C dates that have been excluded, and their values. We mentioned Figure 4 in the text to support our explanation, and we also indicated in Table 2 biased $^{14}$C dates by a star.
Here below is the modified text:
Line 110 to 115: "Nevertheless, because of the low quantity of biogenic carbonates in the proximal part of the Rhone prodelta, we experienced difficulties in dating core RHS-KS55 and observed three age inversions from 336 and 122 cm. Based on seismic and lithofacies correlations at the regional scale (Fanget et al., 2014), we excluded these three 14C dates for core RHS-KS55. We also excluded the three older 14C dates (unit U500, base of the core) since measured ages are considered older than true ages (due to incorporation of reworked benthic foraminifera; for more information, see section 4.1 and section 5.1)."
Line 150 to 159: "Considering the age of the underlying deposits of seismic unit U400 in this area (i.e. paleo-deltaic complex of the Rhone (ERDC), ca. 10.5 ka cal. BP, Berné et al., 2007), we assume that the three 14C dates obtained at 732, 672 and 512 cm (i.e. 12.7, 13.3 and 12.2

ka cal. BP, respectively) are biased (Fig. 4), because of reworking occurring in shallow water environments. This interpretation is supported by the nature of sediments that composed this unit. Indeed, at the regional scale, seismic unit U500 is made of tempestite deposits (Facies 1, section 4.2.) which mainly contain infra-littoral (i.e. upper shoreface) benthic foraminifera (e.g. Elphidium crispum; see Table 2 and Fanget et al., 2014). These infra-littoral benthic foraminifera are likely reworked by high energy hydrodynamic processes that regularly winnowed the seafloor. The three older ages obtained within seismic unit U500 are thus probably the result of reworking during the transgression of an underlying Younger Dryas/Preboreal delta front. It is likely that infra-littoral benthic foraminifera picked for these three 14C dates are reworked from older deposits (for more details, see section 5.1)."

2 - Please enlarge the characters on your figures. They are definitively too little and remain unreadable at 100%. The figure captions are generally too short. Complete the captions is necessary because some feature are not explained in the legends (see some examples below).

a. The black rectangles in fig 5, 6, 8 and 9 are not explained

b. The different grey bands in the same figures, the hatching bands on the figure 8 are not explained

c. You complete the caption of figure 8 but what you mentioned is not indicated on the figure. Events 2.8 and LIA are not indicated there. And please mark that cold relapses are the black rectangles.

d. Probably if all the figure are proposed in portrait format it will be better.

We modified all the figures and enlarge the characters. We also completed the figure captions and gave more information about black rectangles (i.e. Cold relapses), grey bands (i.e. seismic units), dashed lines (i.e. seismic discontinuities), and hatched rectangles (periods of increased hydrology).

Note than in Figure 8, the 2.8 ka event and the LIA correspond to CR4 and CR6, respectively. We explained it in the legend.

We also changed the format of all the figures. There are all in portrait layout now.

3 - Minor remarks

- Have a look on the order of the references. I think that there are some errors (see for example in Magny citations order)

We checked the references and modified the order when necessary.

- Please add a reference in page 3 line 82 for climate or pace the reference at the end of the sentence. Did the authors cited point the link between changes in sediment fluxes and climate?

We moved the citation at the end of the sentence since authors' manuscripts support this observation.

We look forward to hearing from you and if you have any further queries please do not hesitate to contact me.

Yours sincerely,

Anne-Sophie Fanget

**Définition du style :** Normal

[revised manuscript text omitted]

**black rectangles (left side of the figure).**